# Multiple-instance learning of somatic mutations for the classification of tumour type and the prediction of microsatellite status

Jordan Anaya [1], John-William Sidhom[2,3,4], Faisal Mahmood [5,6,7,8,9] & Alexander S. Baras [1,2,4] ✉

Large-scale genomic data are well suited to analysis by deep learning algorithms. However, for many genomic datasets, labels are at the level of the sample rather than for individual genomic measures. Machine learning models leveraging these datasets generate predictions by using statically encoded measures that are then aggregated at the sample level. Here we show that a single weakly supervised end-to-end multiple-instance-learning model with multi-headed attention can be trained to encode and aggregate the local sequence context or genomic position of somatic mutations, hence allowing for the modelling of the importance of individual measures for sample-level classification and thus providing enhanced explainability. The model solves synthetic tasks that conventional models fail at, and achieves best-in-class performance for the classification of tumour type and for predicting microsatellite status. By improving the performance of tasks that require aggregate information from genomic datasets, multiple-instance deep learning may generate biological insight.

Deep learning has made considerable progress in a range of biological tasks[1]. Yet for genomics data this progress has been limited to predicting features of sequence elements and positions in the genome, such as transcription factor binding, DNAse-I sensitivity and histone-based modifications, or whether the sequence functions as a promoter[2,3]. Making predictions at a higher level, such as at the level of a collection of genomic measures, is complicated by the curse of dimensionality—the high dimensional space makes the data sparse and in general promotes overfitting[4]. Current approaches to this problem include manually reducing the dimensionality through feature selection, dimension reduction techniques such as singular value decomposition, negative matrix factorization and various types of autoencoder, or the use of sparse networks that attempt to reduce the weights of the model[5]. However, reducing the dimensions of the data or capacity of the model may produce suboptimal results.

Regardless of how the features of the individual genomic measures are generated, currently a simple aggregation such as a sum or mean is performed to get to a sample-level vector (representing a set of genomic

[1]Department of Pathology, Johns Hopkins University School of Medicine, Baltimore, MD, USA. [2]The Sidney Kimmel Comprehensive Cancer Center, Johns Hopkins University School of Medicine, Baltimore, MD, USA. [3]Department of Biomedical Engineering, Johns Hopkins University School of Medicine, Baltimore, MD, USA. [4]Bloomberg~Kimmel Institute for Cancer Immunotherapy, Sidney Kimmel Comprehensive Cancer Center, Johns Hopkins University School of Medicine, Baltimore, MD, USA. [5]Department of Pathology, Brigham and Women's Hospital, Harvard Medical School, Boston, MA, USA. [6]Department of Pathology, Massachusetts General Hospital, Harvard Medical School, Boston, MA, USA. [7]Cancer Program, Broad Institute of Harvard and MIT, Cambridge, MA, USA. [8]Cancer Data Science Program, Dana-Farber Cancer Institute, Boston, MA, USA. [9]Harvard Data Science Initiative, Harvard University, Cambridge, MA, USA. ✉e-mail: baras@jhmi.edu

**Fig. 1 | Approaches for sparse genomics data.** Data such as somatic mutations must first be encoded and aggregated at the sample level before making predictions about a sample (defined as a set of mutations). Currently the process of encoding and aggregating is handled separately from making predictions with the sample-level vectors. With attention MIL it is possible to encode, aggregate and make predictions with a single end-to-end model. This allows the model to learn a rich feature space at the instance level while also calculating attention for each instance before aggregation, thereby allowing for model explainability.

measures). Then, a model such as a random forest or neural net is applied to these sample vectors to perform the sample-level machine learning task at hand (Fig. 1). This process essentially weights each genomic measure of the set derived from a given sample equally when in fact it may be that some specific measure(s) are more salient. A more modern attention strategy that dynamically weights genomic measures into sample-level feature vectors may identify these specific measures. Moreover, with current approaches, all of the learning occurs at the sample level, and 'end-to-end' training is not possible, which would allow for novel encoding strategies of genomic measures driven by the machine learning task (Fig. 1).

This weakly supervised problem, where features are learned for individual measures (instances) while supervision occurs at the sample level, is the multiple instance learning (MIL) framework[6–8]. MIL has recently revolutionized the field of computational pathology, allowing researchers to identify cancer subtypes or tissues of origin, or predict survival[9–11]. Additional labels in the field of cancer biology may include the presence or absence of cancer, or response to therapy, and sparse genomic measures may be somatic mutations, circulating DNA fragments, neopeptides, RNA/protein modifications, copy number alterations or methylation sites.

Somatic mutations are a complex but well-studied genomic measure, with much of the biology already understood and ample data to test new models. When constructing features for somatic mutations our current understanding of the biology can easily be brought in, such as utilizing information about genes[12–14] or pathways[5]. However, for a given task it may not always be clear what known biology applies,

and some measures may have uncertain biology. In these cases, we can use the fundamental properties of the measure and allow the model to show us which features are important through attention to specific instances and/or the learned representations of the instances. Some fundamental properties of a somatic mutation are its local sequence context, which has been previously summarized by looking at the neighbouring 5′ and 3′ nucleotides[15,16], and its genomic location, which has been represented as 1 Mb bins[17].

Here we present a tool for performing attention MIL and demonstrate its application to somatic mutation data. We use this model to calculate attention for the fundamental properties of mutations, either local sequence context or the genomic position. Using simulated data we explore various MIL implementations on a range of tasks and compare the proposed approach to conventional machine learning approaches in this area. We then apply the model to tumour classification and learn the salient features of sequence and position while exceeding the performance of the current approaches. Finally, we compare our model to state-of-the-art techniques at determining microsatellite status, and our model performs favourably despite the fact that comparable tools use a priori knowledge specific to the task while the proposed approach does not.

## Results

### Aggregation Tool for Genomic Concept
Current applications of machine learning to mutation data are generally limited to an aggregation of hand-crafted features. Our model

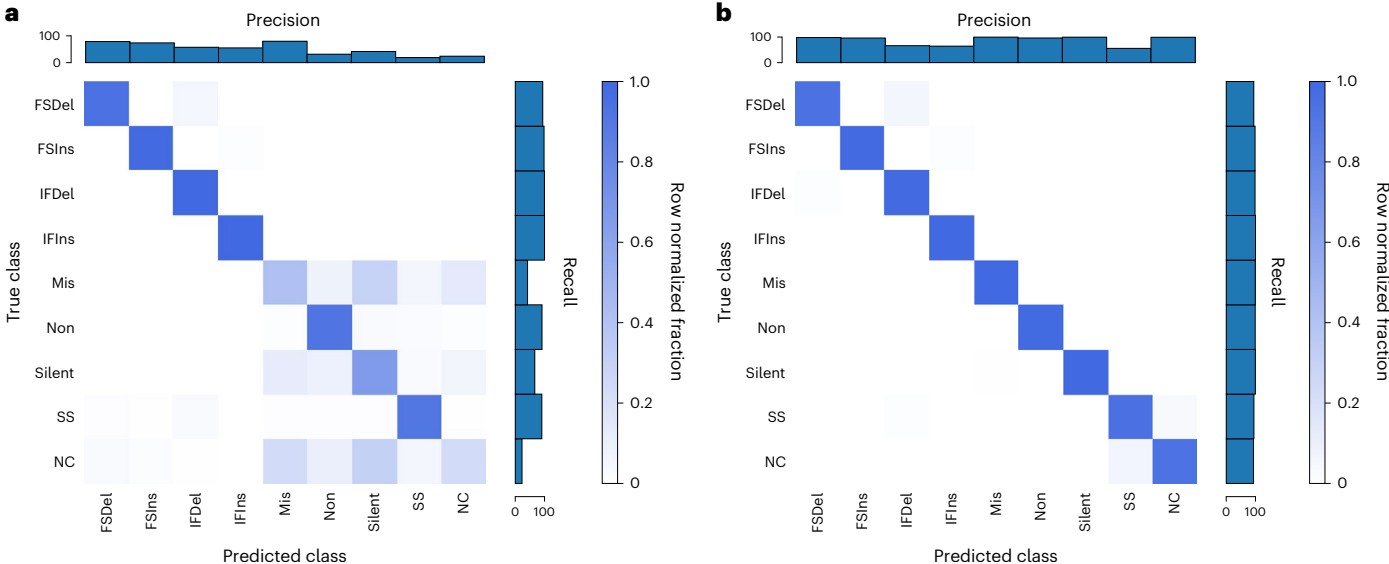

**Fig. 2 | Predicting variant consequence. a,b,** Our sequence encoder can learn variant consequence as defined by variant effect predictor without (**a**) and clearly better with (**b**) reading frame information. FSDel, frameshift deletion;

FSIns, frameshift insertion; IFDel, inframe deletion; IFIns, inframe insertion; Mis, missense; Non, nonsense; SS, splice site; NC, noncoding. All four plots show row normalized confusion matrices.

differs in that attention is first given to individual instances before aggregation, and if desired an end-to-end model is possible allowing for instance features to be learned during training (known as representation learning). Hand-crafted feature engineering may be most efficient when the representation of an instance is well/completely understood and aggregation of information over a set of instances is done via a static function such as sum and mean. Representation learning at the level of the instance can extract features specific to a given machine learning task and, in scenarios with a very large possible set of genomic measures, can be combined with trainable attention mechanisms that can help with model explainability. To allow a model to extract its own features, decisions must be made on how the raw data will be presented to the model and how the model will encode the data. We consider the outcome of this process to be a genomic concept and essential to extracting relevant features. Somatic mutations are often reported in a Variant Call Format or Mutation Annotation Format (MAF), and a genomic concept can be constructed for any measurement in these files. The concept can be as simple as an embedding matrix (for example, our position encoder), or it can be as complex as convolutional layers for the flanking nucleotide sequences along with the reference and alteration in both the forward and reverse directions (our sequence encoder; Fig. 1).

To confirm that the encoders we developed were valid, we performed positive controls, using the unique mutation calls from The Cancer Genome Atlas (TCGA) multi-centre mutation calling in multiple cancer (MC3) public MAF. Our sequence encoder was found to be a faithful representation of a variant, learning the 96 contexts and an outgroup with near-perfect accuracy, and our embedding strategy was able to perform a data compression without any information loss (Supplementary Fig. 1). To confirm that our sequence encoder could effectively utilize strand information, we asked a more difficult question: whether it could classify variants according to their consequence as provided by the MC3 MAF, specifically the consequence/variant classification of frameshift deletion, frameshift insertion, in-frame deletion, in-frame insertion, missense, nonsense, silent, splice site and noncoding (5′ untranslated region, 3′ untranslated region, intron). This problem requires learning all 64 codons in 6 different reading frames, and importantly the strand the variant falls on affects the label. We first asked how well the model could do without providing a reading frame,

and while the model was able to learn insertions and deletions (InDels) and splice sites, it was unable to distinguish noncoding mutations from the other classes (as would be expected) and did the best it could at associating codons with a consequence (Fig. 2a). When provided, the reading frame in the form of strand and coding sequence position modulo 3 (noncoding variants were represented by a zero vector), the sequence concept was now able to correctly classify missense versus nonsense versus silent mutations (Fig. 2b), indicating that the modelling approach is able to learn which strand a feature was on and correctly associate the relevant codons with consequence.

Our implementation of MIL is motivated by ref. 18 (Supplementary Fig. 2), with some important modifications for the nature of somatic mutation data. In image analysis the aggregation function is often a weighted average, but whereas the number of tiles is unrelated to the label for an image the number of mutations may provide information about a tumour sample. Using simulated data, we explored various MIL implementations along with traditional machine learning approaches and found our attention-based MIL with a weighted sum performed well (Supplementary Figs. 3–5). For a weighted sum to be meaningful, the instance features must be activated, and this results in potentially large values on the graph. To account for this, we perform a log of the aggregation on graph. We also developed dropout for MIL, wherein a random subset of instances is given to the model each gradient update, but then all instances are used during evaluation. This can allow for training with large samples and also helps with overfitting as the samples are altered every batch. To improve model explainability, we designed the model for multi-headed attention, where each attention head can be viewed as class-specific attention when the number of attention heads matches the number of classes. The model is implemented in TensorFlow with ragged tensors and is easily extensible to other data types, and we refer to the resulting tool as Aggregation Tool for Genomic Concepts (ATGC).

## Cancer type classification

A readily available label in cancer datasets is the cancer type, and this task can have practical importance for when the tumour of origin for a metastatic cancer cannot be determined[19]. The types of mutation of a cancer are influenced by the mutational processes of its aetiology, while the genomic distribution of its mutations is influenced by

## Table 1 | Tumour classification performance metrics for exome project codes

| Data | Encoding | Aggregation | Model | Accuracy | Weighted accuracy | AUC |
|---|---|---|---|---|---|---|
| 96 contexts | Onehot | Sum | Logistic regression | 46.9% | 47.7% | 0.923 |
| | | | Random forest | 50.2% | 48.8% | 0.925 |
| | | | Neural net | 50.8% | 52.7% | 0.936 |
| | | Weighted sum | ATGC | 51.9% | 54.2% | 0.941 |
| 6 bp windows | Sequence encoder | Weighted sum | ATGC | 58.0% | 59.8% | 0.952 |
| 1Mb bins | Onehot | Sum | Logistic regression | 48.7% | 44.3% | 0.893 |
| | | | Random forest | 44.9% | 38.7% | 0.879 |
| | | | Neural net | 50.4% | 48.3% | 0.913 |
| | Embedding | Weighted sum | ATGC | 53.9% | 51.5% | 0.923 |
| 30 kb bins | Onehot | Sum | Logistic regression | 54.1% | 49.7% | 0.926 |
| | | | Random forest | 47.5% | 40.9% | 0.892 |
| | | | Neural net | NA | NA | NA |
| | Embedding | Weighted sum | ATGC | 56.5% | 53.7% | 0.929 |
| Gene | Onehot | Sum | Logistic regression | 56.9% | 52.4% | 0.932 |
| | | | Random forest | 48.6% | 41.8% | 0.898 |
| | | | Neural net | 55.6% | 52.6% | 0.929 |
| | Embedding | Weighted sum | ATGC | 60.4% | 57.5% | 0.940 |

Every model was trained with the same sample weighting and fivefold cross validation. AUC, area under the curve. NA, when using a large input vector such as the 30 kb bins, our procedure for optimizing neural nets cannot be used.

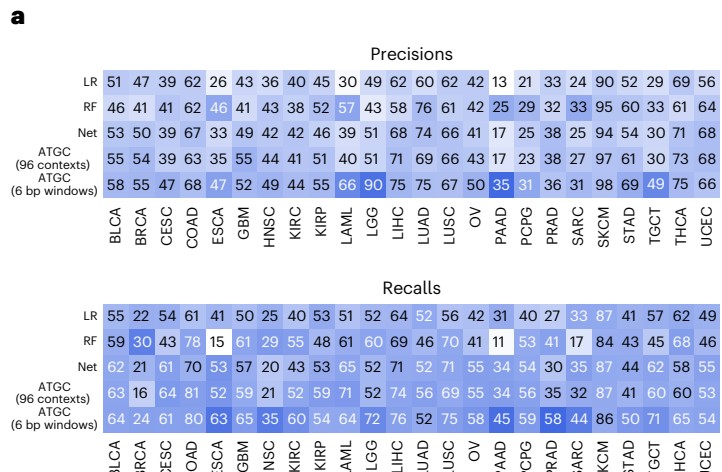

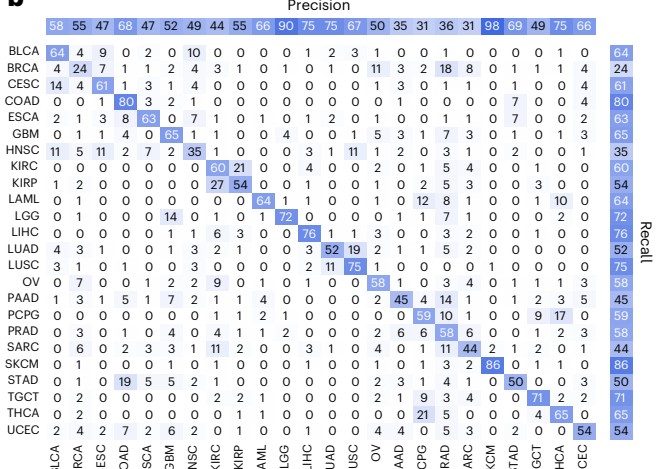

**Fig. 3 | Tumour classification metrics. a**, Precisions and recalls for the models using the 96 contexts and ATGC with the 6 bp windows. **b**, Confusion matrix for ATGC and the 6 bp windows. All numbers are shown as percentages. BLCA, bladder urothelial carcinoma; BRCA, breast invasive carcinoma; CESC, cervical squamous cell carcinoma and endocervical adenocarcinoma; ESCA, oesophageal carcinoma; GBM, glioblastoma multiforme; HNSC, head and neck squamous cell carcinoma; KIRC, kidney renal clear cell carcinoma; KIRP, kidney renal papillary

cell carcinoma; LAML, acute myeloid leukaemia; LGG, brain lower grade glioma; LIHC, liver hepatocellular carcinoma; LUSC, lung squamous cell carcinoma; OV, ovarian serous cystadenocarcinoma; PAAD, pancreatic adenocarcinoma; PCPG, pheochromocytoma and paraganglioma; PRAD, prostate adenocarcinoma; SARC, sarcoma; STAD, stomach adenocarcinoma; TGCT, testicular germ cell tumours; THCA, thyroid carcinoma; UCEC, uterine corpus endometrial carcinoma.

histology of origin, and these features of somatic mutations have been shown to be capable of classifying cancers[17,20–22]. We were interested in seeing how a model that calculates attention or learns it owns features compared to established approaches. For this task we used the TCGA MC3 public mutation calls, which are exome based. To understand the baseline performance on this data using current approaches, we used two common hand-crafted features: the 96 single base substitutions (SBSs) contexts (and a 97th outgroup so that no data were discarded) and an approximately 1 Mb binning of the genome. For each manual feature, we ran a logistic regression, random forest, neural net and our MIL model. We also ran our model with 6 base pair (bp) windows of the sequences and explored gene as an input.

Table 1 shows the results of test folds from fivefold cross validation for the different models and the different inputs for the TCGA project codes (a 24-class problem). For each input our model outperformed current approaches, and the novel input of base-pair windows showed a significant benefit (~14% increase over the best standard model). Notably, our model showed a benefit even when using an identical encoding to the other models (96 contexts), suggesting the attention alone can improve model performance to some degree. To validate these results, we investigated the performance of the 96 contexts in whole genome sequencing and again saw a benefit with our model (86% accuracy compared to 81%; Supplementary Table 1). For additional validation we also ran the models classifying the MC3 samples

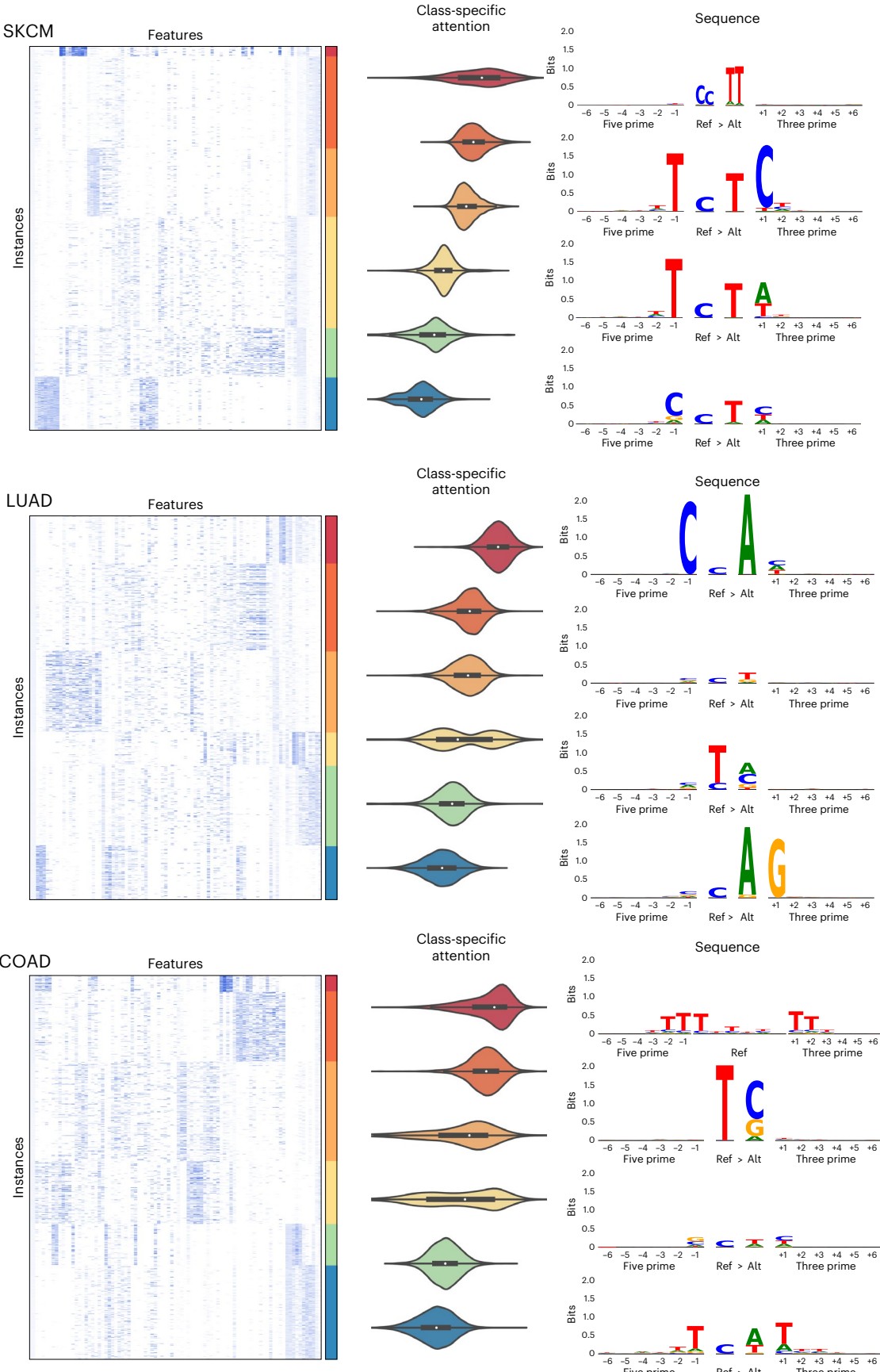

**Fig. 4 | Instance feature vectors reveal known cancer biology.** The feature vectors for the instances of a single test fold for SKCM, LUAD and COAD samples were clustered with K-means into 6 clusters and ordered by their median class-specific attention (2,000 instances sampled at random per cancer shown in the heat maps). Violin plots of the attention values for each cluster are also shown along with sequence logos of the instances for the four clusters with highest median attention.

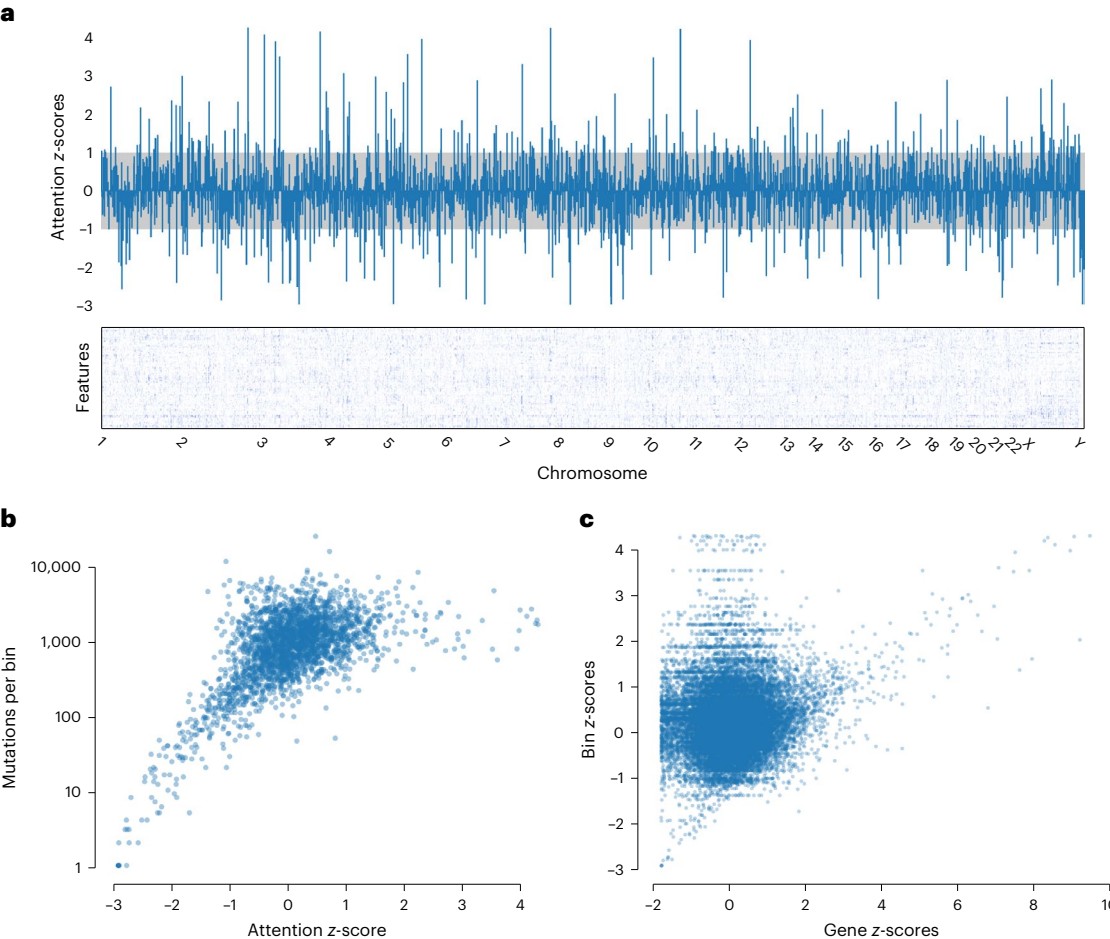

**Fig. 5 | Genomic attention scores. a**, Attention *z*-scores for each 1 Mb bin were calculated across all 24 attention heads and averaged across 5 folds. The embedding matrix for a single fold is shown below with the starts of each chromosome shown. The *z*-scores between −1 and 1 are shaded. **b**, Attention *z*-scores for each 1 Mb bin plotted against the number of mutations falling in each bin. **c**, Attention *z*-scores for each 1 Mb bin plotted against the attention *z*-scores for the corresponding genes in those bins from a model with gene as input.

according to their National Cancer Institute thesaurus (NCIt) codes and again saw a benefit with our model for every input (a 27-class problem; Supplementary Table 2).

To investigate the performance differences, we looked at classification performance in terms of precision and recall for each model stratified by tumour type (Fig. 3a). Our model, which takes 6 bp windows as input, showed significant improvement in oesophageal carcinoma, lower-grade glioma, pancreatic adenocarcinoma and testicular germ cell tumours. The improvement seen in lower-grade glioma is almost certainly due to identifying IDH1 mutations via a mapping of local sequence context of that specific hotspot. We also investigated the predictions of the best performing model (ATGC with 6 bp windows) with a confusion matrix (Fig. 3b) and observed that cancers of similar histologic origin were often mistaken for each other. For the corresponding analyses with gene as input, see Extended Data Fig. 1.

When investigating how the model is interpreting genomic variant data, we can examine both the representation of a variant produced by the model and what degree of attention the model is assigning a variant. To illustrate these two concepts, we show a heat map of the learned variant representation vectors for several cancer types with known aetiologies (Fig. 4). Unsupervised K-means clustering was used to group the instances within each tumour type, revealing a rich feature space for the learned variant representations and clear clusters for each cancer type. We next explored how class-specific attention related to this instance feature space and observed a spectrum of attention

levels. In Fig. 4 we see that for skin cutaneous melanoma (SKCM) six clusters did for the most part produce clusters which were either high or low in attention, while in lung adenocarcinoma (LUAD) and colon adenocarcinoma (COAD) we see clusters that contain a bimodal distribution of attention.

To investigate what sequences were present in each of these clusters, we generated sequence logos. For SKCM the highest attention cluster was composed of a specific doublet base substitution characteristic of ultraviolet radiation, with the next highest attention cluster comprising a very specific 5′ nucleotide, reference (ref), alternative (alt) and 3′ nucleotide also characteristic of ultraviolet radiation[16]. For LUAD the highest attention cluster contained sequences characteristic of tobacco smoking, while in COAD the highest attention cluster was a specific deletion occurring at a homopolymer, which is characteristic of cancers deficient in mismatch repair and is a known signature of this cancer type[16].

While in Fig. 4 we used clustering to identify groups of mutations within a cancer type which may be of interest, we instead could have simply sorted all instances by attention. To explore this possibility, we took the highest attention instances (top 5%) of each attention head for SBSs and InDels and looked at the bits of information contained in the logos (Supplementary Fig. 6). For SBS mutations most of the information gain occurs at the alteration and flanking nucleotides; however, for most heads there is still information two, three or more nucleotides away from the alteration. Given that a head of attention

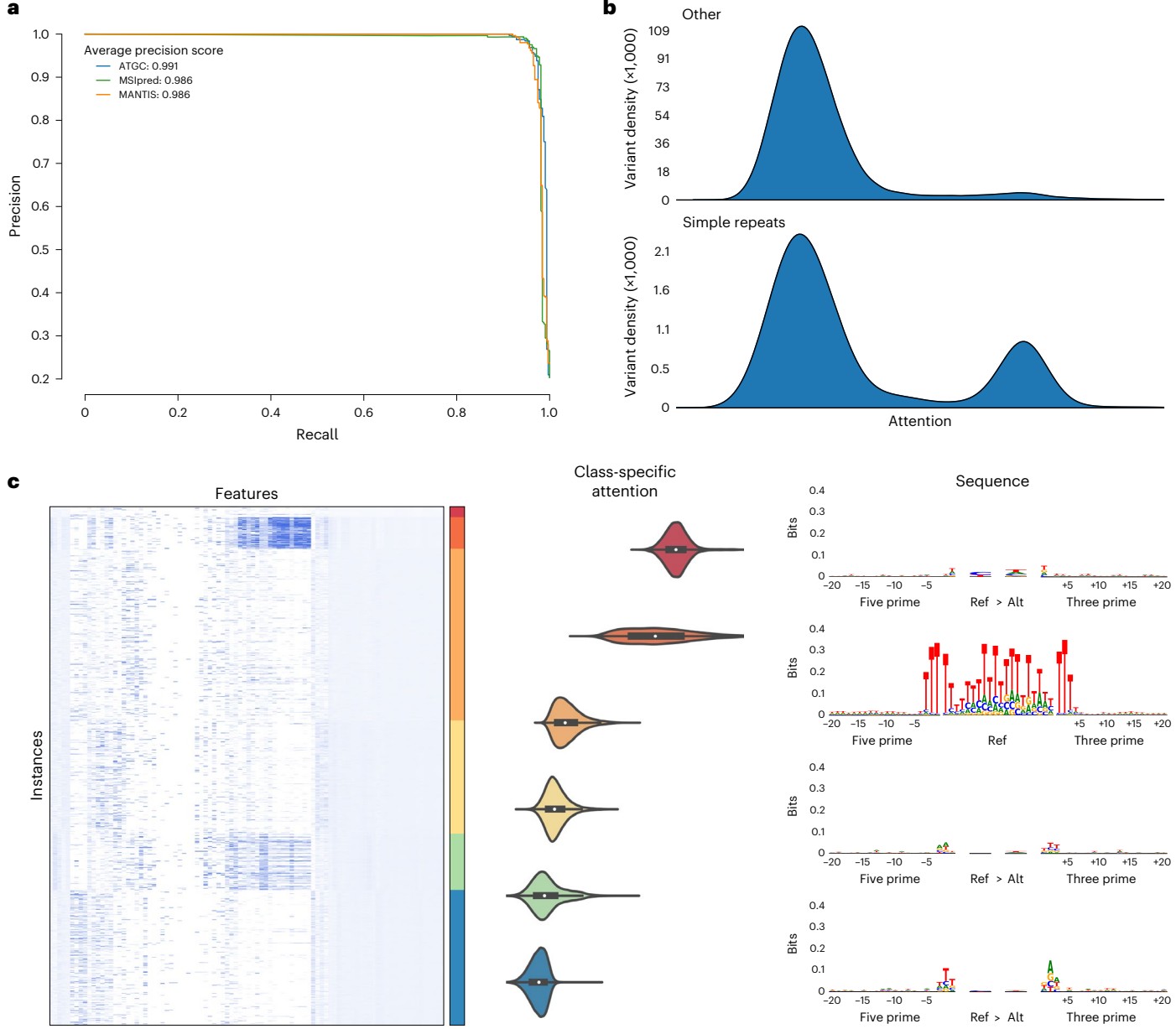

**Fig. 6 | Application of ATGC MIL framework accurately predicts MSI status by learning which are the salient variants. a**, Average precision recall curves over nine stratified K-fold splits for MSIpred and ATGC, precision recall curve for MANTIS for samples which had a MANTIS score. **b**, Probability density plots of the output of the attention layer stratified by whether a variant was considered to be in a simple repeat from the UCSC genome annotations. **c**, Features of 2,000 random instances from a single test fold clustered with K-means and the corresponding sequence logos for the 4 clusters with highest median attention.

may identify several distinct motifs as important, the high attention instances for each head may not be homogeneous, and as a result these bits represent a lower bound of the information gain. For InDels there is significant information three or four nucleotides away from the alteration, and these sequences are often mononucleotide repeats. When performing a similar analysis for our gene encoder, we noticed that a small number of genes appear to be given high attention in each head, and clear groupings were present in the embedding matrix, with a small cluster enriched in cancer-associated genes (Supplementary Fig. 7).

Through the attention mechanism we also investigated what our model learned about genomic location. When looking at the attention values for the 1 Mb bins, the different heads of attention appeared to be giving attention to the same bins, so we calculated the z-scores for each head and averaged across heads. We also noticed the values appeared consistent across folds of the data, so we also averaged over

the five data folds. Figure 5a shows the averaged attention values for the bins across the genome along with an embedding matrix from one of data folds. Specific bins are clearly being either upweighted or downweighted. Investigation of bins with low attention scores revealed that they contained very little data, which caused us to wonder whether attention simply correlated with amount of data in each bin. There is initially a strong correlation between attention and number of mutations in a bin (Fig. 5b), but once bins contain a certain amount of data the relationship disappears. Given that this is exomic data, we suspected the bins with the highest attention contained genes important for cancer classification, so we calculated average gene attention z-scores with our model that used gene as input and matched genes with their corresponding bin (averaging when a gene was split across bins). There is nearly an order of magnitude more genes than bins, so most bins contain multiple genes, and the same attention will be assigned

to all genes in the bin. For the genes with the highest attention there is a corresponding bin also being given high attention (Fig. 5c, top right corner), but there are also many genes that are passengers in those bins and mistakenly being given attention (Fig. 5c, top left corner). This likely explains why using genes as input outperformed models using position bins as input.

## Microsatellite instability

Characterized by deficiencies in the mismatch repair proteins (MLH1, MSH2, MSH6, PMS2), microsatellite unstable tumours accumulate InDels at microsatellites due to polymerase slippage. As with tumour classification, current bioinformatic approaches to predicting microsatellite instability (MSI) status rely on manually featurizing variants. For example, the Microsatellite Analysis for Normal Tumor InStability (MANTIS) tool uses a binary alignment map file to calculate the average difference between lengths of the reference and alternative alleles at predefined loci known a priori to be important[23]. Similarly, the MSIpred tool calculates a 22-feature vector using information from a MAF file and loci again known to be important, specifically simple repeat sequences[24]. Both of these approaches are valid and produce accurate predictions, but they rely on knowing the nature of the problem.

Because mutations characteristic of MSI occur at many different repeat sites, and because the repeats have a distinctive sequence, we chose to use our sequence concept for this problem. For the data we opted to use the controlled TCGA MC3 MAF rather than the public MAF as the public MAF excludes most variants in non-exonic regions and most simple repeats fall in these regions. The TCGA has ground truth labels as defined by PCR assay for some tumour types, and we were able to obtain labels for uterine corpus endometrial carcinoma (494), stomach adenocarcinoma (437), COAD (365), rectum adenocarcinoma (126), oesophageal carcinoma (87) and uterine carcinosarcoma (56) tumour samples. For the sequence concept we went out to 20 nucleotides for each component (5′, 3′, ref and alt) to allow the model to potentially capture long repetitive sequences.

Although we did not provide the model information about cancer type or perform any sample weighting, our model showed similar performance across cancer types (Extended Data Fig. 2a). We believe MANTIS and MSIpred are considered state of the art when it comes to MSI classification performance, and as can be seen in Fig. 6a our model slightly outperforms them despite not being given information about simple repeats. When comparing our predictions to MANTIS, both models were often similarly confident in their predictions (Extended Data Fig. 2b); however, there are cases where MANTIS predicts the correct label but our model does not, or our model predicts the correct label but MANTIS does not, suggesting that the best MSI predictor may be one that incorporates predictions from multiple models. There are a few cases where the sample is labelled MSI high by the PCR, but both MANTIS and our model are confident the sample is MSI low, perhaps indicating the PCR label may not be 100% specific (or alternatively indicates an issue with the binary alignment map files for these samples).

To classify whether a variant falls in a repeat region or not, MSIpred relies on a table generated by the University of California, Santa Cruz (UCSC, http://hgdownload.cse.ucsc.edu/goldenPath/hg38/database/simpleRepeat.txt.gz). Using this file, we labelled variants according to whether they fell in these regions, and as seen in Fig. 6b variants which occurred in a simple repeat region were much more likely to receive high attention than variants that did not. When clustering the instances of these samples, we observed two clear clusters that are receiving high attention: a smaller cluster characterized by SBS mutations in almost exclusively intergenic regions with almost 33% labelled as a simple repeat and a larger cluster characterized by deletions in genic but noncoding regions. The sequence logo of the intergenic cluster did not reveal a specific sequence, while the logo for the deletion cluster revealed that the model is giving attention to deletions at a mononucleotide T repeat.

## Discussion

Many genomic technologies generate data that can be considered 'large p (features), small n (samples)', wherein the number of possible measures/features per sample greatly exceeds the number of samples. For example, somatic mutations can occur anywhere in the genome, thus creating an enumerable number of possible unique features per sample. Similar considerations apply to circulating DNA fragments, CHIP-SEQ peaks, methylation sites or RNA/protein modifications. Attention MIL is a natural solution to these problems because it essentially transposes the problem−the large amount of instance data is a benefit instead of a hindrance when extracting relevant features.

When performing cancer classification, application of our model led to improvement regardless of the input, and when classifying samples according to their MSI status our model slightly outperformed the current state-of-the-art methods despite the other methods containing information specific to the task. Importantly, whereas other models require post hoc analyses to understand how they are making decisions, our tool directly reveals which instances it views as important. This is essential because as these measurements begin to be used in the clinic and researchers turn to deep learning for their analysis, it will likely be necessary for the models to explain their decisions given a patient's right to understanding treatment decisions[25].

Research into how to best implement attention MIL is an active field, with recent activity in its application to computational pathology[26]. We consider our approach to attention MIL fairly standard[18], with our goal being to simply demonstrate its value in the context of genomics data. As such, it is unlikely our results represent the full potential of applying MIL to these data. For example, in computational pathology there have been recent suggestions to include a clustering step at the instance level[9,27,28]. And by design we limited our instance featurization to the factors that uniquely define a mutation: chromosome, position and reference/alternative alleles. We can easily imagine incorporating outside knowledge into the variant encoding process, such as variant consequence, biological pathway(s) involved and so on. We hope that the success we achieved with our proof-of-concept application of MIL inspires additional work in this area.

## Methods
### Model

We used a combination of TensorFlow version 2.7.0 and tf.keras for implementing ATGC. Keras, similar to many deep learning libraries, requires the first dimension of the inputs to match the first dimension of the outputs. Many implementations of MIL work around this constraint by performing stochastic gradient descent, where one sample is shown to the model at a time. This precludes the ability to easily perform sample weighting. To perform minibatch gradient descent as we would with any other model, we developed our model around ragged tensors.

Our model is modular in the sense that the top aggregation model which calculates attention and generates predictions takes as input encoder models. These encoders perform operations on ragged data for instance data and normal vectors for sample data (the dataset functions automatically infer whether a ragged tensor needs to be made). This framework allows for any number of instance or sample inputs, any number of outputs and any number of attention heads (as graphics processing unit (GPU) memory permits). Ragged tensors are fully supported by TensorFlow, so it is possible to use our model with default loss functions and dataset batching. However, because we like additional control over the sample weighting and want the ability to perform stratified batching and data augmentation (data dropout), we prefer to use our own loss and metric classes even when the loss and/or metric already exists in TensorFlow, and we created dataset utilities built around generators.

Our attention strategy was inspired by that proposed in ref. 18. One issue with the proposed attention in MIL is that its value is a function of not only an instance's importance but also its rarity. If two instances

are equally important, but one is present at a much lower fraction (low witness rate), then the rarer instance will be given a higher attention, and if a key instance is very frequent it may be given very little attention if any. In addition, it is possible for the model to find a solution that is characterized by the absence rather than presence of key instances, which results in key instances being given a lower attention. To correct for this issue and make the attention clearer, we added L1 regularization on the output of the attention layer. The more regularization added, the clearer the separation between key instances and negative instances but at the risk of decreased model performance. Another potential issue with attention is that it is independent of the bag. It may be the case that a key instance is only a key instance when it occurs in a certain bag environment, but the model will not make this distinction at the level of the attention layer. If desired, the attention can be made dynamic[29] by sending information about the samples back to the instances (Supplementary Fig. 2c), which may provide additional information about how the model is making decisions. We developed our own version of dynamic attention which calculates a weighted mean with standard MIL attention, then sends that sample vector back to the instances, calculates a second round of attention and performs a second aggregation.

We provide users several options for the aggregation function (mean, sum, dynamic). When using an aggregation function that includes a sum, the instance features should be activated, and this can result in potentially large sums. To counteract this, we log the aggregations on the graph when a sum is performed. Depending on the data there may be too many mutations to fit onto the graph, so we developed dropout for attention MIL where a random fraction of instances per batch are sent into the model during training, but then all the instances are used for evaluation. This has the additional benefit of helping with overfitting, as the samples are constantly changing during training, and is essentially a form of data augmentation.

### Custom activation functions
Functions can be tested at: https://www.desmos.com/calculator/jvwuzpadvd

### Adaptive square root.

$$\mathrm{ASR}(x, \alpha) = \sqrt{e^\alpha + x^2}$$

The adaptive square root (ASR) will be used as a core element in the activation functions below.

### Adaptive rectifying unit.

$$\mathrm{ARU}(x, \alpha) = 0.5 \times (x + \mathrm{ASR}(x, \alpha))$$

Note that a bias term could be added and applied to $x$ before this activation function (as is the convention) and $\alpha$ can be a trainable parameter that modulates the curvature of this 'rectifier'. The adaptive rectifying unit (ARU) can approach the shape of the commonly used rectified linear unit (ReLU) as $\alpha$ approaches negative infinity, but ARU is fully differentiable across all $x$, which does not hold for ReLU. We have tuned default initial conditions for $\alpha$ to match the commonly used softplus function, particularly near $x = 0$, as initial conditions for this adaptive rectifier.

### Adaptive sigmoid unit.

$$\mathrm{ASU}(x, \alpha_{\mathrm{lower}}, \alpha_{\mathrm{upper}}) = \frac{x + \mathrm{ASR}(x, \alpha_{\mathrm{lower}})}{\mathrm{ASR}(x, \alpha_{\mathrm{lower}}) + \mathrm{ASR}(x, \alpha_{\mathrm{upper}})}$$

Again, a bias term could be added and applied to $x$ before this activation function (as is the convention). By design, the function adaptive

sigmoid unit (ASU) is bounded by 0 and 1, at negative and positive infinity (respectively), as is the case with the generic sigmoid function. The lower and upper $\alpha$ parameters control the curvature of this function as it approaches the lower and upper bounds. We have tuned default values to match the generic sigmoid function, particularly near $x = 0$, as initial conditions for this adaptive sigmoid function.

The above were motivated by the previously described inverse square root unit[30]. It is important to note that all input $x$ is at most squared in this formulation and there is no application of input $x$ as an exponent $e^x$ which greatly helps to avoid numerical overflow that can occur in the context of aggregation over samples in a MIL framework.

### MAF processing
For the MC3 public MAF, variants that had a 'FILTER' value of either 'PASS', 'wga' or 'native_wga_mix' and fell within the coordinates of the corresponding coverage WIGs were retained. For the MC3 controlled MAF, variants that had a 'FILTER' value of 'PASS', 'NonExonic', 'wga', 'bitgt', 'broad_PoN_v2', 'native_wga_mix' or combination thereof and were called by more than one mutation caller were retained. The MC3 working group was inconsistent in its merging of consecutive SBSs, so these were merged into a single mutation if the maximum difference between the average alternative or reference counts and any single alternative or reference count was less than 5 or the maximum percentage difference was less than 5%, or the maximum variant allele fraction deviation was less than 5%.

### Tumour classification analyses
We performed tumour classification with both the original project codes and an NCIt ontology for exomic data and with histology codes for whole genome data. For the project codes, we required each class to have 125 samples, resulting in 24 classes and 10,012 samples. For the NCIt classification, tumours were mapped to their NCIt code by A. Baras, a board-certified pathologist, using the available histology description. We required at least 100 samples per NCIt code, resulting in 27 classes and 8,910 samples. For the Pan-Cancer Analysis of Whole Genomes (PCAWG) data only white-listed samples were used, and a donor was only allowed to have a single sample in the dataset. When a donor had more than one sample, the following preference order of specimen type was used for selection of the sample: primary tumour−solid tissue, primary tumour−other, primary tumour−lymph node, primary tumour−blood derived (peripheral blood), metastatic tumour−metastasis local to lymph node, metastatic tumour−lymph node, metastatic tumour−metastasis to distant location, primary tumour−blood derived (bone marrow). To prevent potential issues with GPU memory, we only used samples with less than 200,000 mutations. Any histology with at least 33 samples was used for analysis, resulting in 24 histologies and 2,374 samples. All models were weighted by tumour type, and the K-folds were stratified by tumour type.

**Logistic regressions.** 'LogisticRegression' from 'sklearn.linear_model' (version 1.0.2) was used with default parameters.

**Random forests.** We used 'RandomForestClassifier' from 'sklearn.ensemble' (version 1.0.2), and we explored how the different parameters affected performance with 'gp_minimize' from 'scikit-optimize' (version 0.9.0). We found most parameters to have limited effect; however, we did find the default number of estimators was far from optimal, and set 'n_estimators' to 900 and 'min_samples_split' to 10.

**Neural nets.** The process from the PCAWG consortium was used for constructing the neural nets[17]. Essentially for each fold of the data, 200 different sets of hyperparameters are searched through. As we performed weighted training, we selected the best performing hyperparameters based on weighted cross entropy instead of accuracy. When using gene as input, the default search space caused some issues with

GPU memory and as a result were adjusted, but otherwise the search space was copied exactly.

**MIL models.** Details for each individual MIL model that was run can be found in the accompanying GitHub repository.

## MSI analyses

The PCR labels were merged and when the 5-marker call and 7-marker call were in disagreement, the 7-marker call was given preference. Both microsatellite stable and microsatellite instability low MSI-L were considered to be MSI low. The source code for MSIpred was altered to allow it to run on Python 3, and the model was changed to output probabilities but otherwise was run as recommended. Models were not weighted by tumour type, but tumour type was used for K-fold stratification. For our MIL model, because there were only two classes, we considered it a binary classification task and used a single head of attention. A data dropout of 0.4 was used, and the sequence components were given 8 independent kernels with ARU activation and fused to a dimension of 128 with a ReLU activation and 0.01 L2 kernel regularization; a dropout of 0.5 was performed, then attention was calculated with a single layer and ASU activation and 0.05 L1 activity regularization to force the positive class to receive higher attention. Following aggregation, a layer of 256 and ReLU activation was used followed by 0.5 dropout, a layer of 128 and ReLU activation followed by 0.5 dropout, to the final prediction and no activation.

## Sequence logos

Logomaker[31] (version 0.8) was used for the sequence logos. The probabilities were calculated separately for SBSs, doublet base substitutions and InDels.

## Reporting summary

Further information on research design is available in the Nature Portfolio Reporting Summary linked to this article.

## Data availability

All data used in this publication are publicly available. The MC3 MAFs are from ref. 32, and the MSI PCR labels are available from the cBioPortal, TCGAbiolinks or individual publications[33–36]. The UCSC simpleRepeat.txt was downloaded from http://hgdownload.cse.ucsc.edu/goldenPath/hg19/database/simpleRepeat.txt.gz. The Broad coverage WIGs are available at https://www.synapse.org/#!Synapse:syn21785741. MANTIS values are from ref. 37. The MAF for the ICGC PCAWG samples was obtained from https://dcc.icgc.org/releases/PCAWG/consensus_snv_indel, and the MAF for the TCGA PCAWG samples was obtained from https://icgc.bionimbus.org/files/0e8a845d-a4f4-40bc-890b-5472702d087c.

## Code availability

All code for processing data, running models and generating the figures is available from GitHub at https://github.com/BarasLab/ATGC/tree/method_paper and has also been archived at Zenodo (https://doi.org/10.5281/zenodo.8083498). Code is written in Python 3 and TensorFlow 2. All intersections were performed with PyRanges[38]. We leveraged NVIDIA V100s with 32 GB of RAM for much of the computation; however, most of the computations here could be reasonably performed on CPU as well, within the same coding framework.

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

## Acknowledgements

The results of this study are in whole or in part based on data generated by the TCGA Research Network. J.A. and A.S.B. disclose support for the research described in this study from the Mark Foundation for Cancer Research (19-035-ASP). A.S.B. discloses support for the research described in this study from The Leon Troper, M.D. Professorship in Computational Pathology at Johns Hopkins. F.M. discloses support for the research described in this study from National Institute of General Medical Sciences R35GM138216 and from Brigham and Women's Hospital Presidents Fund.

## Author contributions

J.A., J.-W.S. and A.S.B. conceived the study. J.A. and A.S.B. designed experiments. J.A. performed experiments and analysed results. F.M. and A.S.B. supervised the research. J.A. and A.S.B. wrote the paper with all authors providing input.

## Competing interests

The authors declare no competing interests.

## Additional information

**Extended data** is available for this paper at https://doi.org/10.1038/s41551-023-01120-3.

**Correspondence and requests for materials** should be addressed to Alexander S. Baras.

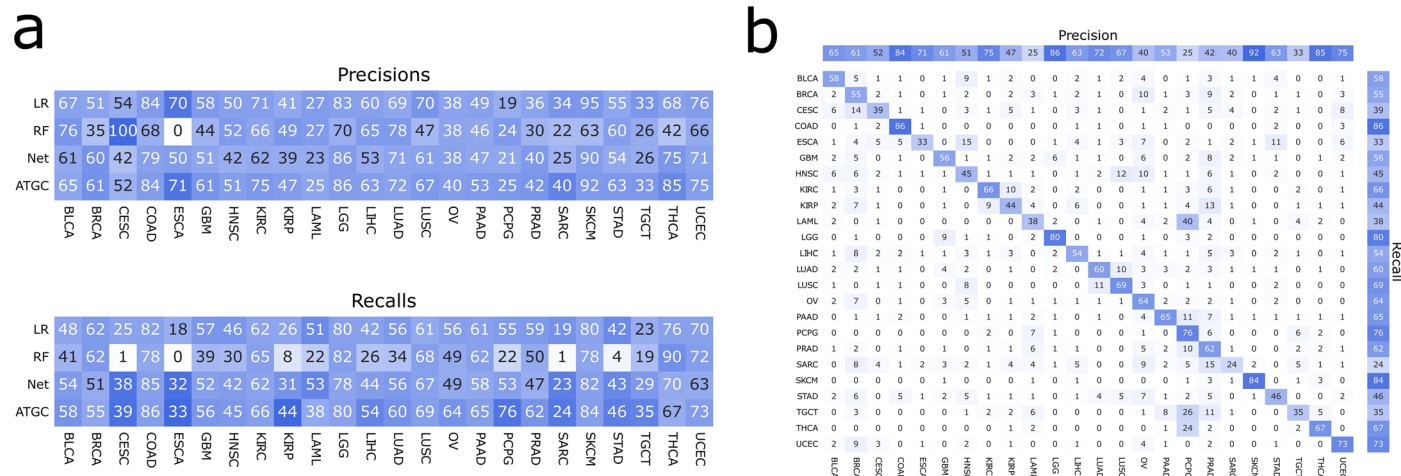

**Extended Data Fig. 1 | Tumour classification metrics. a**, Precisions and recalls for the models using gene as input. **b**, Confusion matrix for ATGC. All numbers are displayed as percents.

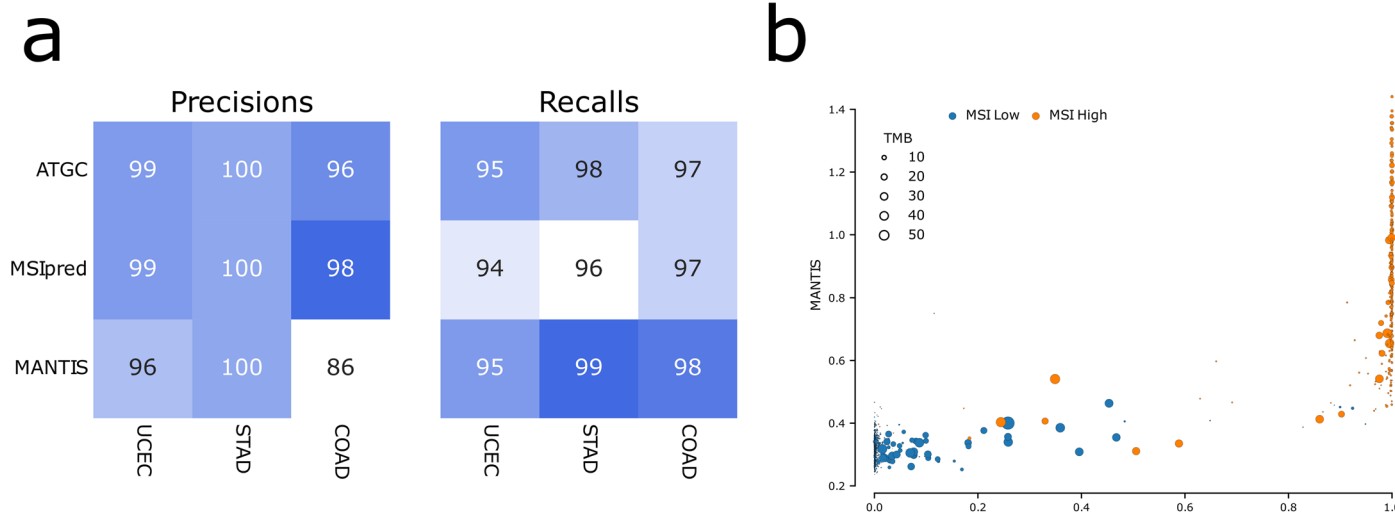

**Extended Data Fig. 2 | MSI predictions across cancer types and between models. a**, Per cancer precisions and recalls for the 3 different models for the 3 most abundant cancer types. **b**, MANTIS scores plotted against ATGC output probability showing high concordance of ATGC model output to MANTIS scores. Samples are colour coded by the PCR-based MSI status label, and the size of each sample corresponds to its total mutational burden (in thousands).

# Reporting Summary

## Statistics

For all statistical analyses, confirm that the following items are present in the figure legend, table legend, main text, or Methods section.

| n/a | Confirmed | |
|---|---|---|
| ☒ | ☐ | The exact sample size (*n*) for each experimental group/condition, given as a discrete number and unit of measurement |
| ☒ | ☐ | A statement on whether measurements were taken from distinct samples or whether the same sample was measured repeatedly |
| ☒ | ☐ | The statistical test(s) used AND whether they are one- or two-sided<br>*Only common tests should be described solely by name; describe more complex techniques in the Methods section.* |
| ☒ | ☐ | A description of all covariates tested |
| ☒ | ☐ | A description of any assumptions or corrections, such as tests of normality and adjustment for multiple comparisons |
| ☒ | ☐ | A full description of the statistical parameters including central tendency (e.g. means) or other basic estimates (e.g. regression coefficient) AND variation (e.g. standard deviation) or associated estimates of uncertainty (e.g. confidence intervals) |
| ☒ | ☐ | For null hypothesis testing, the test statistic (e.g. *F*, *t*, *r*) with confidence intervals, effect sizes, degrees of freedom and *P* value noted<br>*Give P values as exact values whenever suitable.* |
| ☒ | ☐ | For Bayesian analysis, information on the choice of priors and Markov chain Monte Carlo settings |
| ☒ | ☐ | For hierarchical and complex designs, identification of the appropriate level for tests and full reporting of outcomes |
| ☒ | ☐ | Estimates of effect sizes (e.g. Cohen's *d*, Pearson's *r*), indicating how they were calculated |

*Our web collection on statistics for biologists contains articles on many of the points above.*

## Software and code

Policy information about availability of computer code

| Data collection | No new data collection was performed. |
|---|---|
| Data analysis | Python modules used: logomaker==0.8, matplotlib==3.5.1, numpy==1.22.2, pandas==1.4.1, scikit-learn==1.0.2, tensorflow==2.7.0, biopython==1.78, pyranges==0.0.115, scikit-optimize==0.9.0.<br><br>The code that we developed and used in this study is available at https://doi.org/10.5281/zenodo.8083498. |

For manuscripts utilizing custom algorithms or software that are central to the research but not yet described in published literature, software must be made available to editors and reviewers. We strongly encourage code deposition in a community repository (e.g. GitHub). See the Nature Portfolio guidelines for submitting code & software for further information.

## Data

Policy information about availability of data

All manuscripts must include a data availability statement. This statement should provide the following information, where applicable:
- Accession codes, unique identifiers, or web links for publicly available datasets
- A description of any restrictions on data availability
- For clinical datasets or third party data, please ensure that the statement adheres to our policy

All data used in this publication are publicly available. The MC3 MAFs are from ref. 32, and the MSI PCR labels are available from the cBioPortal, TCGAbiolinks or

## Research involving human participants, their data, or biological material

Policy information about studies with underline{human participants or human data}. See also policy information about underline{sex, gender (identity/presentation), and sexual orientation} and underline{race, ethnicity and racism}.

| | |
|---|---|
| Reporting on sex and gender | No sex or gender analyses were performed. |
| Reporting on race, ethnicity, or other socially relevant groupings | No such analyses were performed. |
| Population characteristics | Data from the PCAWG and MC3 working groups were used in this study, with patients from the ICGC and TCGA cohorts. The patient recruitment is described in 'Pan-cancer analysis of whole genomes' (Nature, 2020), and in 'Cell-of-Origin Patterns Dominate the Molecular Classification of 10,000 Tumors from 33 Types of Cancer' (Cell, 2018). |
| Recruitment | Not applicable. |
| Ethics oversight | Not applicable. |

Note that full information on the approval of the study protocol must also be provided in the manuscript.

# Field-specific reporting

Please select the one below that is the best fit for your research. If you are not sure, read the appropriate sections before making your selection.

☒ Life sciences ☐ Behavioural & social sciences ☐ Ecological, evolutionary & environmental sciences

For a reference copy of the document with all sections, see nature.com/documents/nr-reporting-summary-flat.pdf

# Life sciences study design

All studies must disclose on these points even when the disclosure is negative.

| | |
|---|---|
| Sample size | For tumour classification, we set a minimal sample size, the selection of which was arbitrary, but was influenced by the need to have enough samples for stratified K-fold training. |
| Data exclusions | Variants which did not fall within the corresponding coverage WIGs were excluded. |
| Replication | In the case of MSI, that analysis was performed across three different iterations of our model over the last 3 years, along with slightly different approaches to data processing, and the results were highly similar each time. In the case of cancer classification we performed the classification with both project codes and NCIt labels, achieving highly similar results. |
| Randomization | Sklearn was used for randomization, with stratification. |
| Blinding | Not applicable. |

# Reporting for specific materials, systems and methods

We require information from authors about some types of materials, experimental systems and methods used in many studies. Here, indicate whether each material, system or method listed is relevant to your study. If you are not sure if a list item applies to your research, read the appropriate section before selecting a response.

## Materials & experimental systems

| n/a | Involved in the study |
|---|---|
| ☒ | Antibodies |
| ☒ | Eukaryotic cell lines |
| ☒ | Palaeontology and archaeology |
| ☒ | Animals and other organisms |
| ☒ | Clinical data |
| ☒ | Dual use research of concern |
| ☒ | Plants |

## Methods

| n/a | Involved in the study |
|---|---|
| ☒ | ChIP-seq |
| ☒ | Flow cytometry |
| ☒ | MRI-based neuroimaging |

