## [Peer Review File · Nature Biomedical Engineering]

Multiple-instance learning of somatic mutations for the classification of tumour type and the prediction of microsatellite status

Corresponding author: Alexander Baras

Editorial note

This document includes relevant written communications between the manuscript's corresponding author and the editor and reviewers of the manuscript during peer review. It includes decision letters relaying any editorial points and peer-review reports, and the authors' replies to these (under 'Rebuttal' headings). The editorial decisions are signed by the manuscript's handling editor, yet the editorial team and ultimately the journal's Chief Editor share responsibility for all decisions.

Any relevant documents attached to the decision letters are referred to as **Appendix #**, and can be found appended to this document. Any information deemed confidential has been redacted or removed. Earlier versions of the manuscript are not published, yet the originally submitted version may be available as a preprint. Because of editorial edits and changes during peer review, the published title of the paper and the title mentioned in below correspondence may differ.

Correspondence

Tue 09 May 2023

Decision on Article NBME-23-0515A

Dear Dr Baras,

Thank you again for submitting to *Nature Biomedical Engineering* your manuscript, "Aggregation Tool for Genomic Concepts (ATGC): A deep learning framework for somatic mutations and other sparse genomic measures". As noted in previous e-mail correspondence, the manuscript has been seen by three experts, and below you will find the full set of reports (including those of Reviewers #1 and #2, which I had already forwarded to you).

You will see that the reviewers appreciate the work. However, they express concerns about the degree of performance improvement of your multiple-instance-learning-based model. We hope that with substantial further work you can address the criticisms and convince the reviewers of the merits of the study. In particular, we would expect that a revised version of the manuscript provides:

- * Extended benchmarking against relevant published machine-learning classifiers, for the tasks of tumour classification and of the prediction of microsatellite status, as suggested by all reviewers.
- * Enhanced description of the differences and of any advantages of using multiple-instance featurization with respect to other representation-learning strategies also relying on encoding somatic mutations by location and sequence context.
- * Thorough methodological descriptions, as per the relevant comments of all reviewers.

When you are ready to resubmit your manuscript, please upload the revised files, a point-by-point rebuttal to the comments from all reviewers, the reporting summary, and a cover letter that explains the main improvements included in the revision and responds to any points highlighted in this decision.Please follow the following recommendations:

- * Clearly highlight any amendments to the text and figures to help the reviewers and editors find and understand the changes (yet keep in mind that excessive marking can hinder readability).
- * If you and your co-authors disagree with a criticism, provide the arguments to the reviewer (optionally, indicate the relevant points in the cover letter).
- * If a criticism or suggestion is not addressed, please indicate so in the rebuttal to the reviewer comments and explain the reason(s).
- * Consider including responses to any criticisms raised by more than one reviewer at the beginning of the rebuttal, in a section addressed to all reviewers.
- * The rebuttal should include the reviewer comments in point-by-point format (please note that we provide all reviewers with the reports as they appear at the end of this message).
- * Provide the rebuttal to the reviewer comments and the cover letter as separate files.

We hope that you will be able to resubmit the manuscript within 12 weeks from the receipt of this message. If this is the case, you will be protected against potential scooping. Otherwise, we will be happy to consider a revised manuscript as long as the significance of the work is not compromised by work published elsewhere or accepted for publication at *Nature Biomedical Engineering*.

We hope that you will find the referee reports helpful when revising the work. Please do not hesitate to contact me should you have any questions.

Best wishes,

Pep

Pep Pàmies
Chief Editor, Nature Biomedical Engineering

Reviewer #1 (Report for the authors (Required)):

The authors proposed a deep learning method, ATGC, to have the prediction on somatic mutations. They utilized a general strategy in medical imaging analysis, called multiple instance learning, where the input sample is represented as a set of multiple instances instead of a single instance due to high dimensions of features. The authors applied ATGC on simulated data with several tasks, and benchmarked the performance with other conventional machine learning approaches. In addition, ATGC was applied to tumor classification and determining microsatellite status, where ATGC also achieved better performance against current approaches.

I have several major concerns about this paper as the following:

1. Since the input of ATGC is “nucleotide sequences along with the reference and alteration in both the forward and reverse directions”. I would like to know more about the reason why reference and alteration sequences of forward directions are not enough to be the input data.
2. The authors should compare the conventional machine learning methods with different hyperparameters to optimize their performance, which would make the benchmarking fair.
3. To my knowledge, prediction of somatic mutation is a classic problem which is solved by lots of well-

known methods, such as MuTect2, MuSE, VarScan2 and SomaticSniper. These methods could also use MAF file as input. If the authors would like to claim the superiority of performance against state-of-the-art methods, they have to compare these mentioned methods instead of comparing naïve conventional machine learning methods.

4. In the experiment of cancer type classification, why authors choose the number of nucleotide windows as 6 instead of other numbers? Would the different choices of this number affect the performance? The same problem is from results of Microsatellite instability, why 20 is set as the length of nucleotides for each component? How about 30 and 40?

5. Since ATGC tried to solve the multi-class classification problem, how does it solve the data imbalance issue, a common situation in machine learning, which may possibly exist in such a large number of classes from the TCGA dataset?

6. From Supplemental Table 1 and 2, ATGC seems to have not too much superior of performance against neural net, with only 1-2% of accuracy and weight accuracy, and around 0.002 of AUC. I'm pretty curious about the robustness of ATGC and hope authors could prove more about it.

Some minor issues are as follows:

1. Please enlarge the font size of legend and axis from Figure 5A and 5C (sequence).
2. It is not appropriate to claim "We use deep learning techniques to generate features from...". Features could be extracted or detected instead of generated.

Reviewer #2 (Report for the authors (Required)):

"Aggregation Tool for Genomic Concepts (ATGC)"

This manuscript presents a very novel and timely study on the development of a TensorFlow-based tool for weakly supervised deep learning, applied to genetic data obtained from human tumor samples. This is a conceptually new approach that has not been previously attempted for genomic data, making the work both interesting and relevant.

However, I have a few suggestions that I believe would strengthen the manuscript and make it even more impactful:

In the abstract, I recommend providing more specific results, such as the tumor types studied, the number of patients, the prediction tasks, and the performance achieved. This would give readers a better understanding of the scope and outcomes of the actual research, in addition to stating the ideas.

The concept of "weakly supervised" learning is not explicitly mentioned in your manuscript. I suggest defining this term and discussing its implications for your approach in more detail.

In Table 1, you show that your method slightly outperforms other methods, but the performance remains underwhelming. Could you provide additional context or explanations for these results? Are there any potential improvements or optimizations that could be made to your method to enhance performance?

In the introduction, you discuss mutation signatures, but later you do not present any data related to this. I recommend exploring the possibility of predicting mutational signatures, such as those recently proposed by Florian Markowitz's team in Nature, using your new approach. This would provide additional insights into the utility of your method and its potential applications.

It would be interesting to see a comparison or discussion of your approach with potentially more powerful transformer-based approaches. This would help to contextualize your work within the broader landscape of deep learning techniques in genomics.

Please italicize gene names throughout your manuscript, as is standard in the field.

The use of TCGA cohort names such as LGG is not standard in the field. I recommend either defining these

terms or writing them in their full form for clarity.

As the research community is increasingly adopting PyTorch over TensorFlow, it would be beneficial for you to discuss the reasons for your choice of TensorFlow and address any potential concerns or limitations related to this choice.

Reviewer #3 (Report for the authors (Required)):

The authors propose a multiple-instance learning (MIL) framework for classification, called ATGC, in which deep learning is adopted for learning sequence features. They demonstrate the application of the method in three tasks, namely variant classification, tumor classification and MSI status prediction. The performance of ATGC is compared with existing methods. The proposed method has limited novelty and its performance does not stand out, which are detailed below together with my other concerns.

Major:

1, The authors state that “Current applications of machine learning to mutation data are limited to hand-crafted features. Our model differs in that the instance features are learned during training (known as representation learning)”. In fact, using deep learning or representation learning to learn features of variants is not new, and has been utilized in various methods, with examples including “[1] Predicting effects of noncoding variants with deep learning–based sequence model, *Nature Method*, 2015; [2] Deep learning: new computational modelling techniques for genomics, *Nature Review Genetics*, 2019; [3] Mutation-Attention (MuAt): deep representation learning of somatic mutations for tumour typing and subtyping, *bioRxiv*, 2022 ”.

2, The state-of-the-art methods are not used for comparison in the experiments of cancer classification and MSI prediction. For cancer classification, logistic regression and random forests are traditional and general-purpose approaches, and the neural network model is designed by the authors themselves. In addition, the improvement in terms of AUC over the compared methods is marginal. There are recently published methods for cancer classification, which are however not compared. Example papers include [1] A deep learning system accurately classifies primary and metastatic cancers using passenger mutation patterns, *Nature Communications*, 2021; [2] Mutation-Attention (MuAt): deep representation learning of somatic mutations for tumour typing and subtyping, *bioRxiv*, 2022. For MSI prediction, the authors compared the proposed approach to MANTIS (2016) and MSIpred (2018). The recent methods such as MSIFinder (2021) and MIAmS (2019) are not compared. In addition, deep learning-based methods with image as input features are recently developed (such as MSI-net, *Lancet Oncology*, 2021), which should be at least discussed.

For variant consequence classification, because variant consequence is annotated by looking at how variants could change sequence. Is it necessary to use a machine learning approach to perform the classification? Actually, the precision for some type of variants say splice sites is low. The motivation to use machine learning methods to classify variant consequence needs to be strengthened.

3, Some parameters seem to be chosen arbitrarily. For example, on the sequence length, the authors test 6 nucleotides in cancer type classification and 20 nucleotides in MSI prediction. How is the length selected? No experimental results are shown on the influence of sequence length of the prediction performance. For cancer type classification, K-means is used to group instances into 6 clusters. It is unclear how the number of clusters is determined.

4, On the algorithm description, there are multiple places that need to be clarified; otherwise, it is difficult to understand. First, about the encoder (Figure 1), what is the objective function/loss function of the encoder? How is the strand importance calculated? How is the reading frame information used? It is unclear what module 3 (below figure 1) is. Second, a major component of the proposed method is MIL. It is unclear whether (1) sequence features are first learned independent of MIL and then taken as input for MIL or (2) the proposed method is an end-to-end method so that sequence features are learned through optimizing the loss of sample-level predictions in MIL. If it is the former case, the relevance of learned features with downstream tasks is questioned. If it is the latter case, why not show MIL in the main schematic (Figure 1)?

Tue 11 Jul 2023

Decision on Article NBME-23-0515B

Dear Dr Baras,

Thank you for your revised manuscript, "Aggregation Tool for Genomic Concepts (ATGC): A deep learning framework for somatic mutations and other sparse genomic measures". Having consulted with the original reviewers (whose comments you will find at the end of this message), I am pleased to write that we shall be happy to publish the manuscript in *Nature Biomedical Engineering*.

Although Reviewer #3 is not fully satisfied with your replies to their earlier comments, for this work we have placed editorial emphasis on the value of the strategy of attention-based encoding of somatic mutations by genomic location and sequence context, and we appreciate the additional benchmarking after hyperparameter optimization that is included in the revised manuscript. We don't feel that additional benchmarking is thus required in this case. We feel similarly for the apparent request of additional data with different sizes for the sequence context window. I have communicated these points to the reviewer.

We will be performing detailed checks on your manuscript, and in due course will send you a checklist detailing our editorial and formatting requirements. You will need to follow these instructions before you upload the final manuscript files.

Best wishes,

Pep

Pep Pàmies
Chief Editor, Nature Biomedical Engineering

Reviewer #1 (Report for the authors (Required)):

The authors have addressed my comments.

Reviewer #2 (Report for the authors (Required)):

The authors have responded extensively to all of my requests. The revised manuscript is much improved compared to the original version. I have no further requests.

Reviewer #3 (Report for the authors (Required)):

My previous comments have not been thoroughly addressed by the authors. The major issue is that the performance of the proposed method is not well validated (see below).

1, For cancer classification and MSI status prediction, the performance of the proposed method is not compared with state-of-the-art (SOTA) methods, as described in my previous comments. Without comparison to SOTA models, the advantage of the proposed method is questioned.

2, Regarding the size of sequence context, the authors' response is that "The length will depend on the motifs you are interested in". No experiments on investigating the influence of the sequence size are provided.

3, In addition, I think all comments from reviewers should be included in the rebuttal. Currently, the summary comments (i.e. the first paragraph of my previous comments) are missing.

4, Minor issues. The figures in the change-marked version of the manuscript do not show up. In "Cancer type

classification" section, "it own features"->" its own features".

Rebuttal 1

Size of sequence context window

There were some questions regarding the length of the sequences in our sequence encoder. The length will depend on the motifs you are interested in. For example, if you want the sequence encoder to identify hotspot mutations then a specific sequence which is unlikely to occur by chance needs to be learned. With 6 bases to the left and 6 bases to the right there is a $1/4^{12}$ chance of that sequence occurring on one strand. In general longer sequences will allow for more investigation of the surrounding sequence context, but for optimal performance the sequences should likely be just long enough for precise identification of the motifs relevant to the task. This manuscript was meant to show both how effective a sequence encoder can be for different tasks and how it can probe what information surrounding sequences contain (such as by sorting the instances by attention and investigating the bits of information each base pair contains).

REVIEWER #1:

Since the input of ATGC is “nucleotide sequences along with the reference and alteration in both the forward and reverse directions”. I would like to know more about the reason why reference and alteration sequences of forward directions are not enough to be the input data.

You could simply give the model the sequences as they are presented in a MAF, but you run the risk that the model will see a sequence in training in only one direction. When the model encounters that same sequence in the test data in the other direction it will then not recognise it. There are a couple solutions to this problem. When using the 96 contexts researchers use either the pyrimidine or purine as the reference sequence. We could have done something similar (and did experiment with it) where any time the reference in the MAF is not a pyrimidine we would flip the sequence. However, we instead opted to give the model the sequence in both directions, which does double the compute required, but in our testing this data duplication (actually a form of augmentation) performed better than manually flipping the sequences. Our method also allows the model to decide whether it wants to use strand information or not. In most tasks involving mutations the strand likely does not matter, but should strand be important our model will weight the different strands appropriately (a common scenario would be investigating transcriptional strand bias, or perhaps one could imagine a mutation only occurred on a certain strand when a protein was bound and you provided the model CHIP-SEQ information).

The authors should compare the conventional machine learning methods with different hyperparameters to optimize their performance, which would make the benchmarking fair.

With regards to the experiments in the supplement no amount of hyperparameter tuning will allow the logistic regressions or random forests to solve the problem since the problem was defined as learning a specific sequence 5 prime of the alteration and the 96 contexts only include a single nucleotide 5 prime of the alteration. Instead of using the 96 contexts we could have used the 1,536 contexts, which includes 2 nucleotides upstream of the mutation. However to fully solve the problem we need to go 6 nucleotides upstream, which would require the use of the 100,663,296 contexts. We're unsure if the logistic regression or random forest would train with that vector, but the point is that as the sequence you care about gets more and more complex then onehotting all possible sequences is impractical.

We now use the approach from PCAWG¹ for selecting neural net hyperparameters in cancer classification. We had previously used that approach to search for the optimal parameters of the random forests. Depending on the sklearn version the default number of estimators can be 10 for random forests, which we found far from optimal.

To my knowledge, prediction of somatic mutation is a classic problem which is solved by lots of well-known methods, such as MuTect2, MuSE, Varscan2 and SomaticSniper. These methods could also use MAF file as input. If the authors would like to claim the superiority of performance against state-of-the-art methods, they have to compare these mentioned methods instead of comparing naïve conventional machine learning methods.

Somatic mutation detection by algorithms such as those mentioned above in general take in some form of raw genomic sequencing data (such as FASTQ or BAM formats) and output genomic features (such as somatic mutations) in VCF or MAF formats. They in general do not use MAF as input. More important than this detail, our approach has as input genomic features (such as those presented in MAF formats) and uses them to predict something about the sample from which a set of mutations were detected. The ATGC framework we developed in the manuscript is not designed to detect mutations from raw sequencing data like MuTect2, MuSE, Varscan2, etc. For completeness of this response, the deep learning comparators for variant detection would be tools like VarNet and Google's DeepVariant. Rather, our ATGC framework is an extensible deep learning framework that uses the extracted genomic features from such algorithms in the context of attention-based multiple instance learning with a focus on model explainability (particularly in terms of being able to highlight which genomic features were most important in a given machine learning task).

In the experiment of cancer type classification, why authors choose the number of nucleotide windows as 6 instead of other numbers? Would the different choices of this number affect the performance? The same problem is from results of Microsatellite instability, why 20 is set as the length of nucleotides for each component? How about 30 and 40?

We address this above in response comment “Size of sequence context window” above at the top.

Since ATGC tried to solve the multi-class classification problem, how does it solve the data imbalance issue, a common situation in machine learning, which may possibly exist in such a large number of classes from the TCGA dataset?

Other MIL models that we are familiar effectively always use stochastic gradient descent in which the model updates parameters 1 sample at a time. In contrast, because we use ragged tensors of TensorFlow, our model can be trained with minibatching, which is generally considered the optimal way to train neural nets. When sending in 1 sample at a time you cannot easily perform sample reweighting to adjust for class imbalance in the data; in contrast our framework can and does implement sample reweighting to adjust for imbalanced classes.

From Supplemental Table 1 and 2, ATGC seems to have not too much superior of performance against neural net, with only 1-2% of accuracy and weight accuracy, and around 0.002 of AUC. I’m pretty curious about the robustness of ATGC and hope authors could prove more about it.

In this revision of the manuscript we have tried to start from what we believe represents a conventional machine learning approach to the problems being examined and then sequentially add modern attention mechanisms and trainable genomic feature encoding. When using a hand-crafted feature like the 96 contexts, position, or gene we don’t necessarily expect a large boost in performance with a model that calculates attention since it is not extracting any additional information like a novel encoding would. That being said, our new results table does show that the use of a modern attention mechanism to aggregate these data over the context of a set of mutations observed in a given sample does yield an incremental improvement in performance as compared to static aggregation strategies such as a sum. Even if performance had not changed, a new aspect that these attention mechanisms allow for that something like a sum simply cannot is model explainability. It is important to underscore that model performance improvement is not the only aspect of this work (although certainly we would require comparable performance to any other approach). In this revision we highlight this with hand-crafted features pertaining to the global genomic position of a mutation (the genome-wide attention depicted in Figure 5). This approach better allows us to gain insights into the underlying biology without necessarily knowing this upfront. We support this stipulation by showing that a model only given a uniform distribution of bins across the genome is able to highlight important gene locations in the genome (Figure 5).

REVIEWER #2:

The concept of “weakly supervised” learning is not explicitly mentioned in your manuscript. I suggest defining this term and discussing its implications for your approach in more detail.

We have added this term to the manuscript.

In Table 1, you show that your method slightly outperforms other methods, but the performance remains underwhelming. Could you provide additional context or explanations for these results? Are there any potential improvements or optimizations that could be made to your method to enhance performance?

We made a couple optimisations to our models and in cross-validation we now with the 6 bp windows get about a 14% improvement in accuracy over state-of-the-art neural nets that use the 96 contexts in exome sequencing. To get better performance with exomic data it may help to take into account coverage, read depth, and tumour purity of the samples, as those factors affect how many mutations are called, but our goal was to compare our model to current approaches with commonly used inputs and to properties that are intrinsic to the mutation itself (i.e. ref, alt, chr, pos).

Outside of using additional input data, research on how to best perform MIL is an active field, which we feel is somewhat out of scope for this manuscript. We could add an auxiliary task for example, which could be at the instance level or the sample level. You could for example have an instance-level variational autoencoder running at the same time as the MIL model, which may help prevent overfitting, or perhaps add a clustering loss at either the instance or sample level. Our goal with this paper however was not to reinvent the wheel when it came to MIL, but rather to retool a common implementation of attention-based MIL to genomic data such as somatic mutations and investigate its performance and utility. Additionally we developed and openly share an easily extensible tool that researchers can leverage.

In the introduction, you discuss mutation signatures, but later you do not present any data related to this. I recommend exploring the possibility of predicting mutational signatures, such as those recently proposed by Florian Markowitz's team in Nature, using your new approach. This would provide additional insights into the utility of your method and its potential applications.

The mention of mutational signatures in the introduction was meant to introduce the concept of the 96 contexts and how they form the basis of most of the analyses around mutational signatures thus far. A mutational signature is simply a pattern of mutations, with some signatures characterised by very specific mutations. In our discussion of the sequence logos of our cancer classification we mention that the highest attention clusters are characteristic of certain mutational processes, and these correspond to specific signatures in COSMIC. The model explainability in terms of motifs that were driven by the tumor classification task represents a supervised learning approach toward the detection of motifs that show some degree of specificity with respect to tumor type(s).

In order to perform nonnegative matrix factorization (NMF) and extract the different signatures that together define a sample, first the mutations must be given different labels (often the 96 contexts). Our encoder generates a latent representation of the sequence context of a mutation which is essentially a continuous representation of a mutation. We could perhaps cluster all of the latent vectors and then assign each mutation a label based on which cluster it falls in, and then perform NMF with these labels. One potential advantage of this approach is if a supervised task like cancer classification is used then the labels of the mutations themselves will be a reflection of the mutational processes at work.

It would be interesting to see a comparison or discussion of your approach with potentially more powerful transformer-based approaches. This would help to contextualize your work within the broader landscape of deep learning techniques in genomics.

It's unclear if the mention of transformers is in regards to our sequence encoder which performs standard convolutions, or our MIL attention. With regards to our sequence encoder we accurately classified mutations at the instance level and achieved good results at the sample level, but it's always possible there's a better way to represent the sequence of somatic mutations and encode them. The recent use of nucleotide transformers seems to be meant for long-range interactions, and given that we focus on local sequence we're unsure if nucleotide transformers have a role here.

With regards to replacing our attention with the attention mechanism seen in transformer models, the attention mechanism in common transformer architectures² and the attention in multiple instance learning (MIL)³ are similar in that there are values derived from the instances or transformations of those instances and these values are multiplied back against those instances or transformations of those instances. However, the goal of the attention in a transformer is to alter the feature space of each instance given the other instances in a bag (each instance actually becomes a weighted average of the bag) and the goal of attention in MIL is to weight each instance's features for an aggregation step. The motivation for the transformer attention was to allow a model to attend to all the words in a sentence at the same time, since even words far away from each other provide important context, and wasn't designed for an aggregation step. In an MIL problem it is assumed there is no order or dependency of the instances with each other, so there is not a need to calculate all the attention values together, and the attention was developed with aggregation in mind. Essentially with transformer attention each instance becomes a weighted average of the bag, then for a sample-level task these weighted averages would have to again be averaged or summed. With MIL a single weighted average or sum is performed. We believe that problems involving somatic mutations and similar data primarily fall under the MIL framework since mutations are largely independent events that can be featurised without taking into account the other mutations in a bag, and transforming each instance into a weighted average of the bag is likely unnecessary.

The attention mechanism in a transformer is a more complex operation than the attention in MIL since all pairwise similarities are calculated, and this generates a memory intensive matrix if the number of instances is large. An attempt was made recently to utilise the transformer attention for somatic mutations⁴. They encountered this memory issue and were only able to fit 5000 mutations on the graph at a time. The attention in a transformer is also difficult to interpret since each instance now has a vector of attention weights, and for a given mutation these pairwise values change for every single sample. In contrast, the attention in MIL is a single value and does not change based on the bag. The attention in MIL can be potentially made more powerful by allowing it to be multi-headed, and we allow for that in our model and utilise it when possible. This also potentially increases interpretability when the number of heads matches the number of classes. We also have an attention which does depend on the other mutations in a bag which we refer to as dynamic attention, where we essentially concatenate the mean of each bag to each instance, and then calculate attention. In contrast to the attention in a transformer this contextualised attention is not memory intensive.

With that said we do have a separate project that does utilise the transformer architecture to contextualise mutations with respect to their variant allele frequencies (VAFs). While there isn't a clear axis to contextualise the sequences of mutations with respect to each other or where they fall in the genome, knowing the VAF distribution of a sample and where a given mutation falls in that distribution can provide important context to a model for certain problems. We

acknowledge that transformers are an important consideration but would not fit into a manuscript which focuses on “traditional” attention-based MIL. As described above, it should be noted that the transformer architecture does not lend itself as easily and explicitly to classification-specific instance-level attention values as compared to the “multi-headed” attention-based MIL we developed and presented in this manuscript.

Please italicize gene names throughout your manuscript, as is standard in the field.

It seems that the instructions provided by Nature BME have gene names without italics.

The use of TCGA cohort names such as LGG is not standard in the field. I recommend either defining these terms or writing them in their full form for clarity.

When a project code is provided in the text we now write out the full name.

As the research community is increasingly adopting PyTorch over TensorFlow, it would be beneficial for you to discuss the reasons for your choice of TensorFlow and address any potential concerns or limitations related to this choice.

When we started this project 3 years ago TensorFlow was very popular and what our lab was familiar with. Our framework actually has two key components: 1. The use of ragged tensors to allow our model to train just as any other model would, and 2. Dataset utilities that automatically generate ragged tensors for the model to use. Most MIL models we are familiar with are implemented in pytorch and use stochastic gradient descent with a single sample per batch, largely due to the variable number of instances per batch. It is generally considered best practice to perform minibatch training where multiple samples are shown to the model. The ability to send multiple samples in per batch through the use of ragged tensors in tensorflow affords us the ability to leverage a variety of beneficial techniques in machine learning that most MIL formulations would struggle with if having to send only 1 sample per batch. Minibatch training also allows for sample weighting and rank-based losses like that seen in Cox regression. As a result, any MIL model could potentially benefit from using our framework. We are aware of the developments in pytorch with nested tensors, but at this point operations must be explicitly implemented for nested tensors in pytorch. As that continues to evolve and catch up with the more mature ragged tensors of tensorflow we plan to have a pytorch code base for ATGC as well.

Of additional relevance, we leverage the dataset class in tensorflow in conjunction with keras to simplify the generation of batches of ragged tensors for the model training . We also created complex functions to perform data augmentation on a per batch basis, such as performing true MIL dropout.

REVIEWER #3:

The authors state that “Current applications of machine learning to mutation data are limited to hand-crafted features. Our model differs in that the instance features are learned during training (known as representation learning)”. In fact, using deep learning or representation learning to learn features of variants is not new, and has been utilized in various methods, with examples including “[1] Predicting effects of noncoding variants with deep learning-based sequence model, Nature Method, 2015; [2] Deep learning: new computational modelling techniques for genomics, Nature Review Genetics, 2019; [3] Mutation-Attention (MuAt): deep representation learning of somatic mutations for tumour typing and subtyping, bioRxiv, 2022”.

Certainly representation learning has been applied to genomic sequences before, but when it comes to representing a variant (not just a sequence) as it appears in a VCF or MAF (with a reference, alteration, and surrounding context) we are not aware of any models besides ours that encode those features with anything other than onehot vectors such as the 96 contexts. The cited DeepSEA method looks at 1,000 bp regions of the genome to predict chromatin features, which is an instance-level task that does not involve representing a variant (but given this model the authors can then make predictions on the effect of in silico alterations on the chromatin features). We added this reference to the introduction to further distinguish our model from the likes of DeepSEA and others. It should be noted that while we view our sequence encoder as a novel representation of a variant, the true novelty of our work is utilising that encoder in a sample-level task by framing the problem in the context of attention-based MIL, thereby allowing for data driven aggregation of instances to the sample level. We hope our work inspires investigators to consider how they may apply similar thinking to their data, whether that be somatic mutation data or another problem that can be formulated as an MIL task, with a particular focus on the use of an attention mechanism to gain insight.

The cited MuAt work does operate on variants like our model, but uses a variation of the 96 contexts for sequence context, in contrast to the tunable local sequence context window around a mutation that our model allows for. The MuAt work described a sequence feature as “mutation type embedded in a three-nucleotide sequence context”, and clearly state “Sequence contexts, genomic positions and annotations are represented as one-hot encoded vectors.” Given

that this is a static, hand-crafted feature we would not consider that representation learning of mutational sequences.

2, The state-of-the-art methods are not used for comparison in the experiments of cancer classification and MSI prediction. For cancer classification, logistic regression and random forests are traditional and general-purpose approaches, and the neural network model is designed by the authors themselves. In addition, the improvement in terms of AUC over the compared methods is marginal. There are recently published methods for cancer classification, which are however not compared. Example papers include [1] A deep learning system accurately classifies primary and metastatic cancers using passenger mutation patterns, Nature Communications, 2021; [2] Mutation-Attention (MuAt): deep representation learning of somatic mutations for tumour typing and subtyping, bioRxiv, 2022. For MSI prediction, the authors compared the proposed approach to MANTIS (2016) and MSIPred (2018). The recent methods such as MSIFinder (2021) and MIAMs (2019) are not compared. In addition, deep learning-based methods with image as input features are recently developed (such as MSInet, Lancet Oncology, 2021), which should be at least discussed.

We now use the neural net optimisation strategy in [1], and explored additional hyperparameter optimisation for our model. In cross-validation, we now get about a 14% improvement in accuracy over state-of-the-art neural nets from [1] that use the 96 contexts in exome sequencing, which we believe is a significant difference relative to current best alternative model in general for machine learning comparisons. With regards to [2] their approach to featurisation was to find additional ways to onehot information regarding variants, and the only experiment we have in common with their work is cancer classification with the 96 contexts, and in their work they showed lower performance than the neural nets from [1] in this experiment (their Figure 1B, far left bars).

A goal of our work is not necessarily to achieve the best possible results in a given task and specific dataset, but rather to investigate the benefits of applying attention to the fundamental properties of somatic mutations, and demonstrate how an end-to-end model also allows for trainable encoding strategies (such as our local sequence context encoder of variants, including reference and alternative allele). Performance can always be improved by simply adding more and more input features that are relevant to the task, and with somatic mutations this can be tempting since so much fundamental biology is known for the tasks we tackled, but this would not tell us anything about how our model might help a different task with a different and unknown biology.

For MSI classification most models operate on BAM files, and it requires a lot of computational resources to store and process such files. There are many different MSI classifiers that work with BAMs, and applying these tools to the TCGA BAMs is an entire project in itself, for example (Bonneville et al.⁵, or Hause et al.⁶). Comparing MSI predictors that operate on a MAF to those that operate on BAMs isn't a fair comparison, but because the MANTIS values were available for most TCGA samples we went ahead and compared to them anyway. We don't see value in running MSIFinder, MIAMs, or other MSI predictors that utilise BAM files on the TCGA BAMs and comparing to them.

A more fair comparison is another model that operates on MAF files, like MSIPred, and we did run MSIPred on the TCGA data ourselves. Keep in mind that any MSI predictor contains additional information specific to the task and the features are specifically chosen. In contrast, our model just looks at the sequence of all variants and comes up with its own features that it finds relevant. So the fact that our model performs similarly without any information specific to MSI status (and in fact learns this information on its own) is notable. Moreover, the performance of all these tools is very high so any performance benefit will be marginal. The novel aspect of our model is that it is the first to learn its own features for this task and highlight those features with attention.

For variant consequence classification, because variant consequence is annotated by looking at how variants could change sequence. Is it necessary to use a machine learning approach to perform the classification? Actually, the precision for some type of variants say splice sites is low. The motivation to use machine learning methods to classify variant consequence needs to be strengthened.

The variant consequence classification was simply a positive control. We stated in the manuscript on line 78: "To confirm that the encoders we developed were valid we performed positive controls". Because of the nontrivial nature of developing a variant encoder that could dynamically learn the concept of "strandness" we felt it was necessary to document that aspect of the encoder via consequence prediction that would need to dynamically learn this biological concept where appropriate. We would not recommend using our encoder to predict a splice site, it was simply one of the labels available in the MAF file in terms of variant classification.

3, Some parameters seem to be chosen arbitrarily. For example, on the sequence length, the authors test 6 nucleotides in cancer type classification and 20 nucleotides in MSI prediction. How is the length selected? No experimental results are shown on the influence of sequence length of the prediction performance. For cancer type classification, K-means is used to group instances into 6 clusters. It is unclear how the number of clusters is determined.

We address the length of the sequences above in response comment “Size of sequence context window” above at the top. The K-means clusters were purely for visualisation purposes. There are a wide range of clustering methods and techniques for identifying the optimal cluster number. For some cancers the optimal number of clusters may be less than or more than 6. We simply wanted to show the rich feature space of our encoder, and that clustering of the feature space results in clusters with different attention values, and that the high attention clusters have sequences characteristic of known mutational signatures.

4, *On the algorithm description, there are multiple places that need to be clarified; otherwise, it is difficult to understand. First, about the encoder (Figure 1), what is the objective function/loss function of the encoder? How is the strand importance calculated? How is the reading frame information used? It is unclear what module 3 (below figure 1) is. Second, a major component of the proposed method is MIL. It is unclear whether (1) sequence features are first learned independent of MIL and then taken as input for MIL or (2) the proposed method is an end-to-end method so that sequence features are learned through optimizing the loss of sample-level predictions in MIL. If it is the former case, the relevance of learned features with downstream tasks is questioned. If it is the latter case, why not show MIL in the main schematic (Figure 1)?*

If the encoder is being used in the context of a MIL supervised learning task then the loss function is generally defined based on the learning task and in an end-to-end framework the weights of the encoder will be driven by that. In addition, we also presented a couple instance-level “positive control” experiments to ensure our encoder can faithfully represent known features of variants such as learning the 96 trinucleotide contexts and variant consequences. Regarding learning strand importance, we have a custom layer which initially weights each strand equally, but that weighting is trainable and the model can choose to give more weighting to either the forward or reverse strand (which again would be driven by the specific learning task). Reading frame was only used when that information was needed for the problem, such as our test of learning variant consequence. Reading frame was given to the model as the remainder of the CDS position / three (i.e. modulo 3), and was concatenated to strand information along with the variant features. Code for our variant sequence encoder is present at https://github.com/OmnesRes/ATGC/blob/method_paper/model/Sample_MIL.py.

Regarding the use of the sequence encoder in the context of attention-based MIL implemented by our model, our proposed sequence encoder was always trained in an end-to-end mode in which the instance encoder is trained from scratch for the given learning task. We would like to highlight that in this revision we tried to more clearly delineate both whether: (a) a fixed (previous known, i.e. 96 trinucleotide context) encoding scheme was used versus a trainable encoder (b) a fixed MIL aggregation strategy was used (i.e. sum, mean, etc) versus trainable attention-based MIL aggregation that results in a dynamic weighted sum in our implementation. As suggested by the reviewer we have significantly improved Figure 1 to include and delineate both the instance encoding strategies along with MIL aggregation approaches.

REFERENCES

1. Jiao, W. *et al.* A deep learning system accurately classifies primary and metastatic cancers using passenger mutation patterns. *Nature Communications* **11**, 728. ISSN: 2041-1723 (2020).
2. Vaswani, A. *et al.* Attention is all you need. *Advances in neural information processing systems* **30** (2017).
3. Ilse, M., Tomczak, J. M. & Welling, M. Attention-based deep multiple instance learning. *arXiv preprint arXiv:1802.04712* (2018).
4. Sanjaya, P., Waszak, S. M., Stegle, O., Korbelt, J. O. & Pitkanen, E. Mutation-Attention (MuAt): deep representation learning of somatic mutations for tumour typing and subtyping. *bioRxiv* (2022).
5. Bonneville, R. *et al.* Landscape of Microsatellite Instability Across 39 Cancer Types. *JCO Precision Oncology*, 1–15 (Nov. 2017).
6. Hause, R. J., Pritchard, C. C., Shendure, J. & Salipante, S. J. Classification and characterization of microsatellite instability across 18 cancer types. *Nature Medicine* **22**, 1342–1350 (Oct. 2016).